# Fitness Cost of *bla*_NDM-5_-Carrying p3R-IncX3 Plasmids in Wild-Type NDM-Free *Enterobacteriaceae*

**DOI:** 10.3390/microorganisms8030377

**Published:** 2020-03-07

**Authors:** Tengfei Ma, Jiani Fu, Ning Xie, Shizhen Ma, Lei Lei, Weishuai Zhai, Yingbo Shen, Chengtao Sun, Shaolin Wang, Zhangqi Shen, Yang Wang, Timothy R. Walsh, Jianzhong Shen

**Affiliations:** 1Beijing Advanced Innovation Center for Food Nutrition and Human Health, College of Veterinary Medicine, China Agricultural University, Beijing 100193, China; matengfei@cau.edu.cn (T.M.); fujiani_1106@163.com (J.F.); 18744023898@163.com (S.M.); lucia9sea@163.com (L.L.); zws18363976198@163.com (W.Z.); shenyb123@163.com (Y.S.); sunctx@icloud.com (C.S.); shaolinwang@cau.edu.cn (S.W.); szq@cau.edu.cn (Z.S.); WalshTR@cardiff.ac.uk (T.R.W.); 2Department of Medical Microbiology and Infectious Disease, Institute of Infection & Immunity, Heath Park Hospital, Cardiff CF14 4XN, UK

**Keywords:** *bla*_NDM-5_-carrying p3R-IncX3 plasmids, fitness cost, plasmid stability, conjugative transfer, wild-type recipients

## Abstract

The wide dissemination of New Delhi metallo-β-lactamase genes (*bla*_NDM_) has resulted in the treatment failure of most available β-lactam antibiotics, with IncX3-type *bla*_NDM-5_-carrying plasmids recognised as having spread worldwide. In China, bacteria carrying these plasmids are increasingly being detected from diverse samples, including hospitals, communities, livestock and poultry, and the environment, suggesting that IncX3 plasmids are becoming a vital vehicle for *bla*_NDM_ dissemination. To elucidate the fitness cost of these plasmids on the bacterial host, we collected *bla*_NDM_-negative strains from different sources and tested their ability to acquire the *bla*_NDM-5_-harboring p3R-IncX3 plasmid. We then measured changes in antimicrobial susceptibility, growth kinetics, and biofilm formation following plasmid acquisition. Overall, 70.7% (29/41) of our *Enterobacteriaceae* recipients successfully acquired the *bla*_NDM-5_-harboring p3R-IncX3 plasmid. Contrary to previous plasmid burden theory, 75.9% (22/29) of the transconjugates showed little fitness cost as a result of plasmid acquisition, with 6.9% (2/29) of strains exhibiting enhanced growth compared with their respective wild-type strains. Following plasmid acquisition, all transconjugates demonstrated resistance to most β-lactams, while several strains showed enhanced biofilm formation, further complicating treatment and prevention measures. Moreover, the highly virulent *Escherichia coli* sequence type 131 strain that already harbored *mcr*-*1* also demonstrated the ability to acquire the *bla*_NDM-5_-carrying p3R-IncX3 plasmid, resulting in further limited therapeutic options. This low fitness cost may partly explain the rapid global dissemination of *bla*_NDM-5_-harboring IncX3 plasmids. Our study highlights the growing threat of IncX3 plasmids in spreading *bla*_NDM-5_.

## 1. Introduction

The World Health Organization has identified the global spread of antimicrobial resistance as one of the three greatest threats to human health [1,2]. A New Delhi metallo-β-lactamase-1 (NDM-1)-encoding gene was first identified in a clinical *Klebsiella pneumoniae* isolate in 2008 [3]. Since then, the worldwide dissemination of NDM-positive strains has become a serious threat not only to human public health, but also to the poultry industry and the environment [4,5,6,7]. NDM enzymes can hydrolyze most β-lactams, including carbapenems, and bacteria producing these enzymes are associated with a high number of infections [8]. Moreover, *bla*_NDM_ is often associated with other resistance genes, which complicates treatment and results in higher mortality rates [9].

Plasmids are key elements in the dissemination of antibiotic resistance via horizontal transfer. Among the reported *bla*_NDM_-carrying plasmids, IncX3 appears to be the most common type (117/355 = 32.96%) [10]. Interestingly, whole-genome sequencing data obtained from GenBank showed that 68.4% (80/117) of all reported *bla*_NDM_-carrying IncX3 plasmids have been recovered from China, indicating an epidemic clustering of these plasmids. However, *bla*_NDM_-carrying IncX3 plasmid have also been reported in Asia, Europe, and North America [10]. To date, 11 NDM variants, including *bla*_NDM-1_, *bla*_NDM-4_, *bla*_NDM-5_, *bla*_NDM-6_, *bla*_NDM-7_, *bla*_NDM-13_, *bla*_NDM-17_, *bla*_NDM-19_, *bla*_NDM-20_, and *bla*_NDM-21_, have been identified on IncX3 plasmids, indicating that IncX3 plasmids provide an efficient platform for the spread and evolution of *bla*_NDM_ genes and a likely vehicle for the spread of β-lactam resistance [10]. IncX3 plasmids have a narrow host spectrum limited to *Enterobacteriaceae* species and encode a type IV secretion system enabling conjugative transfer, providing accessory functions such as resistance gene acquisition and biofilm formation to their host strains [11,12]. Strains containing *bla*_NDM-5_-carrying IncX3 plasmids have been isolated from clinical settings, companion animals, agriculture, and the environment, indicating that IncX3-type plasmids have played a crucial role in disseminating *bla*_NDM-5_ amongst *Enterobacteriaceae* [7,13,14,15].

However, although plasmids can help their host to adapt to diverse environmental conditions, they also have a fitness cost in the absence of selection for plasmid-encoded traits [16]. Generally, plasmid during bacterial cell division results in reduced growth rate and competitiveness without selection stress [17]. Therefore, the fitness costs associated with *bla*_NDM-5_-carring IncX3 plasmids need to be further explored. A previous study described the physiology and pathogenicity of hyper-virulent *K. pneumoniae* following acquisition of the p24835-NDM5 plasmid [18]. A recent study evaluated the growth kinetics and fitness cost of NDM-1 plasmid carriage in engineered strains *Escherichia coli* J53 and *K. pneumoniae* PRZ [19]; however, that study only examined two strains. Herein, we collected 41 wild-type NDM-free *Enterobacteriaceae* isolates from different sources, including the environment, chickens, meat, flies, humans, swine, and companion animals, to assess the success rate of conjugative transfer of a *bla*_NDM-5_-harboring p3R-IncX3 plasmid. We then evaluated the fitness costs associated with plasmid acquisition. Our study indicated that p3R-IncX3 plasmids are highly stable across multiple generations and can slightly improve membrane formation in different wild-type recipient hosts. The low fitness cost of this plasmid in *Enterobacteriaceae* may contribute to the spread of antibiotic resistance.

## 2. Materials and Methods

### 2.1. Bacterial Strains and Strain Construction

The *bla*_NDM-5_-harboring p3R-IncX3 plasmid used in our study was 46,149 bp in length and had a G+C content of 46.64%. The plasmid had a similar genetic background to that of IncX3 plasmid pNDM_MGR194, originally recovered from a *K. pneumoniae* isolate from India (GenBank accession number KF220657) [14]. The donor strain, *E. coli* 3R (ST156) harboring plasmid p3R-IncX3, was isolated from a chicken cloaca sample from a farm in Qingdao in 2015. PacBio whole-genome sequencing (Sinobiocore, Beijing, China) of strain 3R identified four plasmids with different Inc types (Appendix A), one of which was an IncX3-type plasmid containing *bla*_NDM-5_ and named as p3R-IncX3. Four specific primers were designed to distinguish these four plasmids in strain 3R (Appendix A). To generate a DH5α strain containing the p3R-IncX3 plasmid only, plasmids extracted from *E. coli* 3R were electroporated into *E. coli* strain DH5α (Takara Bio, Kusatsu, Japan). The transformants were first selected using meropenem, the PCR-based screening using the four specific primers was then conducted to identify the transformants harboring the p3R-IncX3 only (Appendix A).

### 2.2. Sampling of Recipients and Conjugative Transfer Assay

Simple random sampling (https://www150.statcan.gc.ca/n1/edu/power-pouvoir/ch13/prob/5214899-eng.htm) was conducted to select *bla*_NDM_-negative strains from our laboratory culture collection for use as wild-type recipients. Briefly, laboratory-stored strains of various sources were given serial numbers, then certain numbers were randomly selected by a computer. During random sampling, each member of a population has an equal chance of being included in the sample. The recipient strains were originally isolated from various sources, including humans, chickens, swine, companion animals, flies, meat, and environment samples. We aimed to select three *E. coli* and three *K. pneumoniae* isolates from each of the different origins; however, if this was not possible, other *Enterobacteriaceae* strains were used as alternatives. Detailed information on the wild-type recipients is available in Appendix A. DH5α harboring the p3R-IncX3 plasmid was used as the donor for conjugation assays. The conjugative transfer assays were performed as previously described [6]. In brief, donor and wild-type recipients were mixed at a ratio of 1:3 on a microporous membrane on Luria–Bertani (LB) agar (Luqiao, Beijing, China) and cultured overnight. The mixtures were collected and then plated on LB agar containing meropenem (2 μg/mL) and chloramphenicol (30 μg/mL), gentamicin (4 μg/mL), or tetracycline (10 μg/mL) depending on the antibiotic susceptibilities of DH5α + p3R-IncX3 and the wild-type recipients (Table 1). The successful recombinant strains were verified by PCR.

### 2.3. Whole-Genome Sequencing

Whole-genome sequencing of all recipient strains, the DH5α donor strain, and their transconjugants was performed using the Illumina HiSeq 2500 sequencing platform (Bionova, Beijing, China). Genomic DNA was extracted using a TIANamp Bacteria DNA Kit (TIANGEN Biotech Company, Beijing, China). The draft genomes were assembled using SPAdes version 3.9.0 [20]. Resistance genes were identified using SRST2 Toolkit version 0.2.0 [21], while multi-locus sequence typing (MLST) and plasmid replicon typing were conducted using online tools (http://www.genomicepidemiology.org/). Minimum spanning trees were constructed for *E. coli* and *K. pneumoniae* using BioNumerics version 7.0. The main phylogroups of the *E. coli* isolates were identified using Clermontyping [22]. The STs, Inc plasmid types, antimicrobial resistance genes, and phylogroups of the isolates were annotated in the neighbor-joining tree using the online tool iTOL (https://itol.embl.de/).

### 2.4. Antimicrobial Susceptibility Testing

The MICs of a range of antibiotics against all of the bacterial isolates were detected using the agar dilution method as per the Clinical and Laboratory Standards Institute recommendations [23] and the EUCAST guidelines (http://www.eucast.org). Reference strain *E. coli* ATCC 25,922 (Tianhe, Hangzhou, China) was used as the quality control strain.

### 2.5. Plasmid Stability

Plasmid stability experiments were performed as described previously [24]. Briefly, wild-type strains carrying p3R-Incx3 plasmid were plated on LB agar (Luqiao, Beijing, China) overnight, then three colonies were randomly selected to passage for 15 days in LB broth medium (Luqiao, Beijing, China) without antibiotic pressure. On the 1st, 3rd, 5th, 7th, 9th, 11th, 13th, and 15th day of passage, samples were diluted and plated on LB agar (Luqiao, Beijing, China) with or without 2 mg/L meropenem. At each time point, 100 colonies of stains were selected for PCR detection (Takara Bio, Kusatsu, Japan) of *bla*_NDM-5_ and the plasmid replicon of p3R-IncX3 to verify the presence of the plasmid. The stability assays were performed in triplicate.

### 2.6. Growth Kinetics

Growth curves for the recipients and transconjugants were performed in 96-well flat-bottom plates (Corning Inc., Corning, NY, USA) as described previously [18]. Briefly, strains were cultured in LB medium (Luqiao, Beijing, China) at 37 °C overnight and then diluted to an optical density at 600 nm (OD_600_) of 0.5. Samples were then diluted 100-fold in LB broth medium (Luqiao, Beijing, China) in 96-well microtiter plates (Corning Inc., Corning, NY, USA) and incubated at 37 °C for 24 h. OD_600_ measurements were taken hourly to construct a growth curve. Growth kinetics assays were performed in triplicate.

### 2.7. Biofilm Formation

Biofilm formation assays were conducted as described previously [25]. Briefly, strains were inoculated into 2-mL tubes containing 1 mL of LB broth (Luqiao, Beijing, China) and then cultured at 37 °C in a shaking incubator overnight. Each culture was then adjusted to a cell density equivalent to a 0.5 McFarland standard. Aliquots (200 μL) of each culture were then added in triplicate to 96-well flat bottom plates (Corning Inc., Corning, NY, USA) and incubated at 37 °C for 48 h. Following incubation, culture was aspirated, and wells were washed twice with 200-μL volumes of PBS. The biofilms were then fixed in methanol for 10 min. After drying, wells were stained with 1% crystal violet solution (Sigma-Aldrich, St Louis, MO, USA) for 10 min, rinsed with PBS until colorless, and dried in a 42 °C incubator. Finally, biofilms were dissolved in 100 μL of 30% acetic acid. Absorbance values were measured at 590 nm using an Infinite M Plex microplate reader (Tecan, Männedorf, Switzerland).

## 3. Results and Discussion

### 3.1. High Success Rates of Conjugative Transfer of the p3R-IncX3 Plasmid

To investigate the conjugative transfer rate of the *bla*_NDM-5_-harboring p3R-IncX3 plasmid, 41 NDM-negative wild-type recipient strains from different sources, including environmental samples, chickens, meat, flies, humans, swine, and companion animals, were randomly selected (Appendix A). *E. coli* 3R (ST156), isolated in our previous study [6], contains four different Inc-type plasmids, one of which is a *bla*_NDM-5_-harboring p3R-IncX3 plasmid. In the current study, we used strain 3R as a donor to generate a *bla*_NDM-5_-harboring p3R-IncX3 plasmid-carrying DH5α strain, which was confirmed by PCR-based screening (Appendix A). Among the 41 wild-type recipient strains conjugated with DH5α-IncX3, 29 strains successfully acquired the p3R-IncX3 plasmid, including 18 *E. coli*, seven *K. pneumoniae*, one *Enterobacter cloacae*, one *Raoultella planticola*, one *Citrobacter freundii*, and one *Enterobacter hormaechei*. This is the first time that *E. hormaechei* has been shown to be able to harbor the *bla*_NDM_-carrying p3R-IncX3 plasmid. The high conjugative transfer rate suggests that p3R-IncX3 plasmids have the potential to increase their host range and further disseminate *bla*_NDM_ genes. A summary of the conjugative transfer results is presented in Appendix A. The overall conjugative transfer rate for *Enterobacteriaceae* from all sources was 70.7%, although the success rate for *E. coli* (78.3%) was higher than rates for *K. pneumoniae* (63.6%) and other species (57.1%) (Figure 1A and Appendix A). Of note, the recipient strains isolated from chicken (71.4%), meat (83.3%), and swine (83.3%) had higher than average (70.7%) success rates (Figure 1B and Appendix A). In comparison, the strains isolated from flies had the lowest success rate (60.0%) (Figure 1B and Appendix A). The high conjugative transfer rates of the p3R-IncX3 plasmid among wild-type *Enterobacteriaceae* isolates from different sources highlight the threat represented by these plasmids.

### 3.2. Phylogenetic Relationships among the Recipient Strains

To investigate the phylogenetic relationships among the recipient strains and to verify the transconjugants, all 41 wild-type recipient strains and 29 transconjugants were subjected to whole-genome sequencing. Phylogenetic trees for the *E. coli* and *K. pneumoniae* strains were constructed from core-genome single-nucleotide polymorphism data, which also included information on the collected strains, including STs, phylogroups, sources, Inc types, and antimicrobial resistance gene profiles (Figure 2). The main differences among the wild-type strains with and without the p3R-IncX3 plasmid were the presence/absence of genes carried by the plasmid. MLST showed that *E. coli* and *K. pneumoniae* strains belonging to 17 and 5 STs, respectively, successfully captured the *bla*_NDM_-carrying p3R-IncX3 plasmid (Appendix A), indicating that the plasmid has a wide host range among the *Enterobacteriaceae*. Phylotype data suggested that more than half of the *E. coli* strains (60.9%, 14/23) belonged to non-pathogenic phylogroup A. Phylogroup B strains composed the second-largest group, which included donor strain 3R. Among the phylogroup B strains, *E. coli* strain No. 28 (ST131), originally isolated from a human, belonged to highly virulent phylogenetic group B2 [26], and could cause severe extraintestinal infections [27]. Significantly, strain No. 28 and the other three strains (Nos. A19, 10194, and 3) all carried *mcr-1*, a plasmid-mediated colistin resistance gene that can breach the efficiency of colistin as a “last-line” resort for treatment of MDR bacterial infections [28]. Therefore, the co-existence of *mcr-1* and *bla*_NDM-5_ in *E. coli* strains, especially the highly virulent ST131 strain, is particularly concerning for public health. Plasmid Inc type analysis showed that two *E. coli* strains both contain an IncX1 plasmid belonging to the same family as the IncX3 plasmid; however, the presence of this plasmid did not affect the ability of these two strains to capture the IncX3 plasmid. The coexistence of different Inc-families in a single strain increases the antimicrobial resistance spectrum of the recipient strains.

### 3.3. Resistance Profiles and Stability of the Transconjugants Carrying the p3R-IncX3 Plasmid

Of the 41 NDM-negative wild-type recipient strains, most were susceptible to meropenem (MIC from 0.032 to 0.125 mg/L, based on the European Committee on Antimicrobial Susceptibility Testing (EUCAST) 2017 guidelines: S, ≤ 2 mg/L; R ≥ 8 mg/L). Strains No. 4 and No. 13 were the exception, with meropenem MIC values of 4 mg/L for both strains. However, all 29 strains that successfully acquired the p3R-IncX3 plasmid demonstrated high-level resistance to meropenem following plasmid acquisition. For example, after acquiring the *bla*_NDM_-carrying p3R-IncX3 plasmid, the MIC of meropenem against DH5α increased from 0.032 to 16 mg/L. For some wild-type recipient strains, the meropenem MICs increased from 0.032 to ≥ 128 mg/L (Table 1). These different increases in meropenem resistance among the recipient strains may be related to the genomic backgrounds of the strains. All transconjugants were also resistant to several β-lactam antibiotics, including ampicillin, ceftriaxone, ceftiofur, and amoxycillin/clavulanic acid, reflecting the fact that NDM-1 shows activity against all β-lactam antibiotics except aztreonam [3,8,29]. Appendix A outlines the changes in MIC of various antibiotics following plasmid acquisition. However, plasmid acquisition only affected the resistance of the transconjugants to β-lactam antibiotics, with no observed changes in resistance to the other tested antibiotics (chloramphenicol, gentamicin, tetracycline, and tigecycline) (Table 1). The expanded antibiotic resistance profiles of the strains following acquisition of the *bla*_NDM_-carrying p3R-IncX3 plasmid confirms the urgent need to implement measures against the spread of resistance genes via IncX3 plasmids.

To evaluate the stability of the p3R-IncX3 plasmid, we selected nine transconjugants (Nos. 1, 7, 14, 20, 26, 28, C45, 0970, and 10194) and passaged them daily for 15 days in the absence of antibiotic selection. Unlike a previous study showing that hyper-virulent *K. pneumoniae* rapidly lost a *bla*_NDM-5_-harboring p3R-IncX3 plasmid after monoculture for 100 generations without antibiotics [18], our results indicated that the transconjugants retained the p3R-IncX3 plasmid, indicating that it was highly stable in the different wild-type strains (Appendix A).

### 3.4. Growth Kinetics and Fitness Cost of the p3R-IncX3 Plasmid

To evaluate the fitness cost of the *bla*_NDM-5_-harboring p3R-IncX3 plasmid, growth kinetics assays were performed for the strains with and without the p3R-IncX3 plasmid. Interestingly, results vary for the different strains. The growth of DH5α carrying the p3R-IncX3 plasmid was almost indistinguishable from that of wild-type DH5α (Figure 3A). Among the environmental strains, all five transconjugants showed equivalent growth rates to those of the parental strains (Figure 3A,B). In comparison, significant decreases in the growth rates of the transconjugants compared with the wild-type strains were observed for chicken-derived strains *E. coli* No. 11 (*p* < 0.01) and *K. pneumoniae* No. 14 (*p* < 0.01) (Figure 3C), chicken meat-derived *C. freundii* strain No. 21 (*p* < 0.01) (Figure 3F), swine-derived *E. coli* strain No. C45 (*p* < 0.01) (Figure 3J), and companion animal-derived *E. coli* strain No. 0970 (*p* < 0.01) (Figure 3K), indicating a fitness cost of keeping the p3R-IncX3 plasmid in these hosts. On the contrary, chicken-derived *E.*
*coli* strain No. 10 (*p* < 0.01) (Figure 3D) and companion animal-derived *E. coli* strain No. 10,194 (*p* < 0.01) (Figure 3L) showed both faster growth rates and higher cell densities at stationary phase as a result of plasmid acquisition. For the remaining strains, no differences in the growth dynamics of the p3R-IncX3 -positive and -negative strains were observed, indicating little or no fitness burden (Figure 3A,B,D,E,G–I). In total, 75.9% of strains showed no change in growth kinetics, 6.9% showed enhanced growth, and 17.2% showed impaired growth as a result of p3R-IncX3 plasmid carriage (Figure 3M). Thus, the growth kinetics assays revealed that the acquisition of p3R-IncX3 plasmid rarely confers a fitness cost to the host.

It is widely believed that in the absence of antibiotic pressure, plasmids containing the corresponding resistance genes will impose certain fitness costs on their hosts [30]. We did not use any antibiotics in our growth kinetics assays. Theoretically, the *bla*_NDM-5_-harboring p3R-IncX3 plasmid should impose a fitness cost on the recipients in the absence of antibiotic pressure as was observed in a previous study wherein a 140-kb *bla*_NDM-1_-carrying IncA/C-type plasmid imposed a high fitness cost in *E. coli* J53 and *K. pneumoniae* PRZ [19]. However, wherever there is oppression, there is resistance. Plasmids will regulate the expression of their own genes through transcriptional regulation to minimize plasmids costs [16]. For example, histone-like nucleoid structuring protein (H-NS) transcriptional repressors can silence the expression of acquired genes [16]. Further, plasmid-encoded H-NS-like genes have been shown to assist plasmid conjugation by reducing the cost of plasmid acquisition [31]. The *bla*_NDM-5_-harboring p3R-IncX3 plasmid used in the current study carries H-NS-like genes, which may explain the low fitness cost of this plasmid. Similarly, two carbapenemase-encoding plasmids, pG12-KPC-2 and pG06-VIM-1, had low biological costs when they transferred from *K. pneumoniae* to *E. coli* [32]. Interestingly, the *bla*_NDM-5_-harboring p3R-IncX3 plasmid used in this study had different fitness costs in different wild-type recipient strains. However, this is consistent with a previous study showing that the fitness costs of newly acquired mobile genetic elements depend on the genetic makeup of the recipient and/or environmental factors [33].

### 3.5. Changes in Biofilm Formation as a Result of p3R-IncX3 Plasmid Carriage

We also compared the ability of biofilm formation in the strains with and without the p3R-IncX3 plasmid. The wild-type *E. coli* DH5α demonstrated only weak biofilm formation, but it increased significantly after getting the p3R-IncX3 plasmid (*p* < 0.05, Figure 4A). Among the 29 wild-type recipient strains, two strains (6.9%) showed decreased ability of biofilm formation following plasmid acquisition (Figure 4I). A significant decrease was observed for *E. cloacae* strain No. 3 (*p* < 0.05) from chicken manure compost and *K. pneumoniae* strain No. 14 (*p* < 0.01) from a chicken cloaca swab (Figure 4B,C). As a result of IncX3 plasmid acquisition, a further nine strains (9/29, 31.0%) showed significantly enhanced ability of biofilm formation (Figure 4I), including *E. coli* and *K. pneumoniae* strains from chickens (No. 13, *p* < 0.01), meat (No. 19, *p* < 0.05 and No. 20, *p* < 0.01), flies (No. 22, *p* < 0.05 and No. 23, *p* < 0.05), humans (No. 28, *p* < 0.05), and swine (A19, *p* < 0.01 and C45, *p* < 0.01) (Figure 4C–G). However, most strains (18/29, 62.1%) showed no significant changes in the ability of biofilm formation (Figure 4I).

Biofilm formation requires the cooperation of different bacterial strains and species, allowing them to prosper and protect each other. Biofilms can enhance the tolerance of bacteria to harsh environmental conditions, and can increase antibiotic resistance [34,35]. In our study, 31.0% of strains exhibited varying increases in biofilm formation ability after obtaining the p3R-IncX3 plasmid. Although most evidence to date suggests that p3R-IncX3 plasmids are narrow host range plasmids of the *Enterobacteriaceae* [9], a growing number of studies show that *bla*_NDM_-carrying p3R-IncX3 plasmids have the capacity to expand their host range among *E. coli* with different MLST profiles [7,12]. Given the broad-spectrum resistance conferred by these plasmids, treatment failure is likely to become more common, especially given that some highly virulent strains, such as ST131 strains carrying *mcr-1*, also demonstrated enhanced biofilm formation following acquisition of the p3R-IncX3 plasmid. In addition, we previously showed that biofilm formation by *Staphylococcus aureus* accelerated the spread of plasmid-borne antibiotic resistance genes via conjugation or mobilization [36]. Thus, the increased biofilm formation ability of strains following acquisition of p3R-IncX3 plasmids may not only accelerate the dissemination of *bla*_NDM_ genes, but also increase the possibility of treatment failure rates.

## 4. Conclusions

This is the first study to confirm high conjugative transfer rates of a *bla*_NDM-5_-harboring p3R-IncX3 plasmid among a wide range of wild-type *Enterobacteriaceae*. The increased resistance to β-lactam antibiotics and enhanced biofilm formation ability as a result of plasmid acquisition may reduce drug efficacy. Further, transfer of this plasmid to highly virulent strains will increase their pathogenicity and multidrug resistance, with the likely consequence of increased treatment failure rates. The low fitness cost of the plasmid further indicated that p3R-IncX3 plasmids may adapt to an expanding range of hosts, thereby disseminating *bla*_NDM_ genes among a large number of strains. These findings highlight the need to recognize the importance of IncX3 plasmids and take measures to control their spread.

## Figures and Tables

**Figure 1 microorganisms-08-00377-f001:**
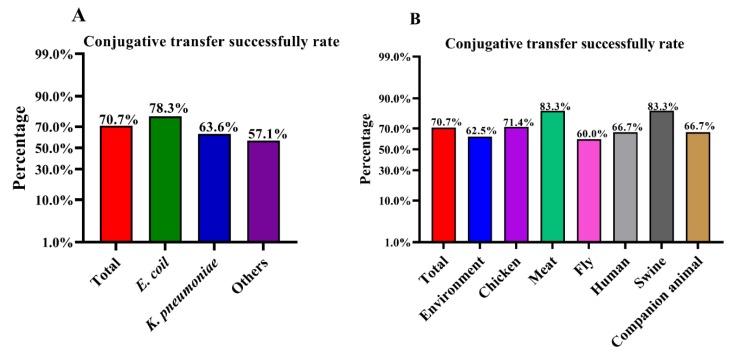
Success rates for conjugative transfer into recipient strains from different sources. (**A**) Overall conjugative transfer success rate and rates among different species. (**B**) Overall conjugative transfer success rate and rates for strains from different sources.

**Figure 2 microorganisms-08-00377-f002:**
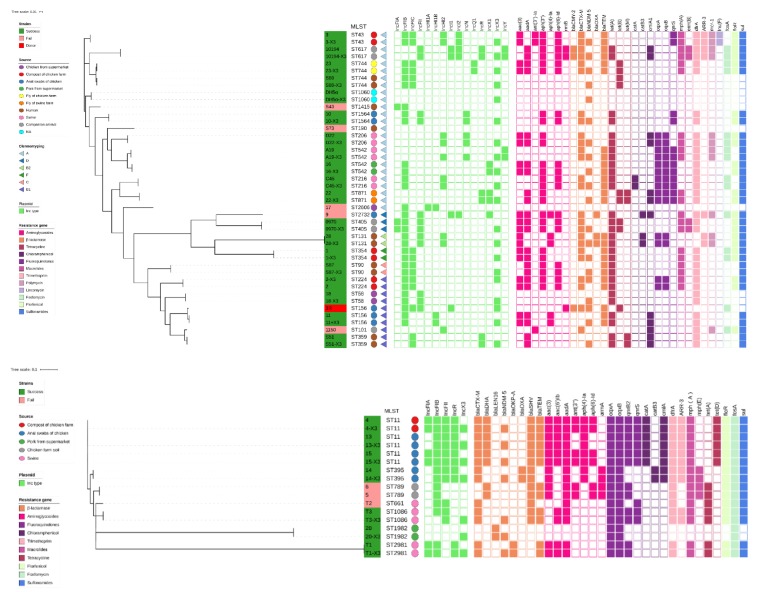
Core genome-based phylogenetic trees for the *Escherichia coli* and *Klebsiella pneumoniae* strains, respectively. Donor strain 3R was used as a reference in the *E. coli* phylogenetic tree. Different background colors in the strain name column indicate the outcome of conjugative transfer (green: success; pink: failure). The main phylogroups are depicted by differently colored triangles. The sources of the isolates are indicated by differently colored circles. Inc types and antimicrobial resistance gene profiles are indicated by the differently colored rectangles (open: negative; filled, positive).

**Figure 3 microorganisms-08-00377-f003:**
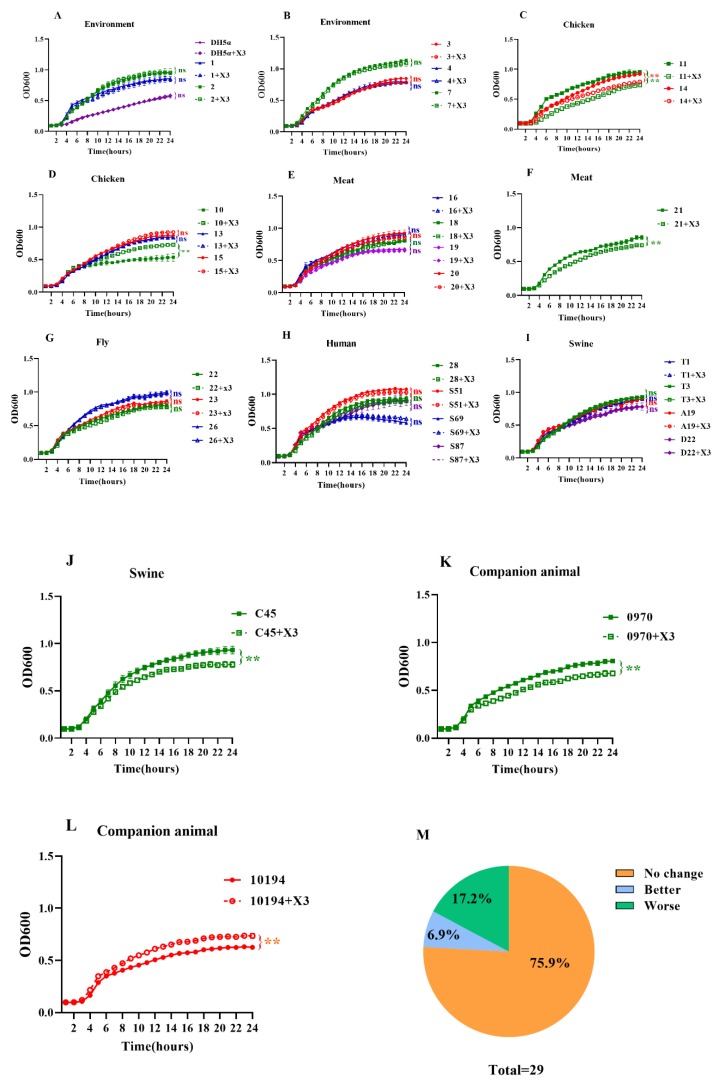
Growth kinetics of strains with and without the p3R-IncX3 plasmid. Strains from different sources and their corresponding transconjugants were examined. DH5α served as the control. Growth kinetics were examined for strains isolated from (**A**) the environment, (**B**) chickens, (**C**) meat, (**D**) flies, (**E**) humans, (**F**) swine, and (**G**) companion animals, (**H**) human, (**I**) swine, (**J**) swine, (**K**) companion animal, (**L**) companion animal. (**M**) Percentages of strains showing no change, enhanced, or impaired growth following acquisition of the p3R-IncX3 plasmid. *, *p* < 0.05; **, *p* < 0.01, ns means no differences.

**Figure 4 microorganisms-08-00377-f004:**
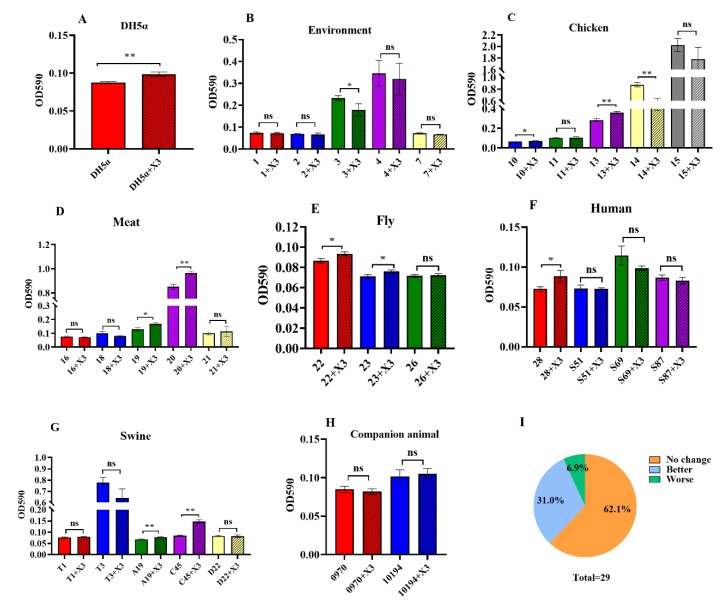
Biofilm formation ability. Biofilm formation ability of strains from different sources, as well as that of their corresponding transconjugants. (**A**) DH5α control. Strains used in the biofilm formation assay were isolated from (**B**) the environment, (**C**) chickens, (**D**) meat, (**E**) flies, (**F**) humans, (**G**) swine, and (**H**) companion animals. (**I**) The percentages of strains showing no change, enhanced, and impaired biofilm formation following acquisition of the p3R-IncX3 plasmid. *, *p* < 0.05; **, *p* < 0.01, ns means no differences.

**Table 1 microorganisms-08-00377-t001:** Antimicrobial susceptibility profiles of recipient strains with and without the p3R-IncX3 plasmid.

No.	Species	Sources	MIC (mg/L) of Antimicrobial Agents																			
			MEM		CT		AMP		CRO		EFT		C		CN		AMC		TET		TGC	
			+	−	+	−	+	−	+	−	+	−	+	−	+	−	+	−	+	−	+	−
DH5α	***E. coli***	TAKARA	16	0.032	0.05	0.05	>512	1	512	0.06	256	0.125	2	2	0.25	0.25	128/64	1/0.5	2	2	0.25	0.25
1	*E. coli*	chicken manure compost	>128	0.032	2	2	>512	1	512	128	>256	64	128	>128	16	16	128/64	16/8	>128	128	1	0.5
2	*E. coli*	chicken manure compost	64	0.06	1	2	>512	>512	>512	512	>256	>256	128	>128	128	>128	>128/64	32/16	64	64	0.5	0.5
3	*E. coli*	chicken manure compost	64	0.06	8	8	>512	>512	>512	512	>256	>256	>128	>128	64	64	128/64	16/8	128	128	1	0.5
4	*K. pneumoniae*	chicken feces	128	4	16	16	>512	>512	512	256	>256	256	>128	>128	>128	>128	>128/64	>128/64	128	128	8	8
7	*E. cloacae*	chicken manure compost	32	0.06	1	1	>512	256	512	0.125	256	1	64	128	0.25	0.25	>128/64	128/64	>128	128	2	0.5
10	*E. coli*	chicken cloaca	64	0.06	4	4	>512	>512	>512	128	>256	128	128	128	0.5	0.5	>128/64	16/8	128	128	0.125	0.125
11	*E. coli*	chicken cloaca	64	0.06	2	2	>512	>512	>512	0.125	>256	1	32	32	8	8	>128/64	32/16	128	128	0.125	0.125
13	*K. pneumoniae*	chicken cloaca	64	4	4	4	>512	>512	>512	256	>256	256	128	128	16	16	>128/64	128/64	>128	>128	0.25	0.25
14	*K. pneumoniae*	chicken cloaca	>128	0.25	1	1	>512	>512	>512	512	>256	256	>128	>128	>128	>128	>128/64	>128/64	32	16	1	0.5
15	*K. pneumoniae*	chicken cloaca	>128	0.125	16	16	>512	>512	>512	256	>256	256	>128	>128	16	16	>128/64	>128/64	>128	>128	8	4
16	*E. coli*	pork, farmer market	128	0.032	1	2	>512	512	512	0.03	>256	0.5	128	>128	64	64	>128/64	16/8	128	128	0.25	0.06
18	*E. coli*	Chicken, supermarket	64	0.032	1	1	>512	32	512	0.06	>256	0.5	4	4	0.5	0.5	>128/64	2/1	128	128	0.5	0.5
19	*R. ornithinolytica*	Chicken, supermarket	128	0.06	2	2	>512	64	512	0.06	256	0.5	1	1	0.25	0.125	>128/64	2/1	16	16	0.5	0.125
20	*K. variicola*	Pork, retail store	32	0.06	1	1	>512	64	>512	0.25	>256	1	4	4	1	1	>128/64	8/4	4	4	2	2
21	*C. freundii*	Chicken, farmer market	16	0.06	1	1	>512	64	512	0.5	256	2	8	8	2	2	>128/64	128/64	>128	>128	0.125	0.125
22	*E. coli*	fly on pig farm	32	0.032	1	1	>512	>512	512	0.25	256	0.25	>128	>128	0.25	1	128/64	8/4	>128	>128	0.25	0.125
23	*E. coli*	fly on chicken farm	64	0.06	1	2	>512	>512	512	128	>256	128	128	64	64	16	128/64	16/8	>128	>128	0.25	0.125
26	*E. hormaechei*	fly on pig farm	64	0.06	1	1	>512	512	512	0.25	256	1	128	>128	2	2	>128/64	128/64	128	128	0.5	0.5
28	*E. coli*	human	128	0.06	8	16	>512	>512	>512	256	>256	256	128	>128	16	16	>128/64	32/16	64	128	0.125	0.125
T1	*K. quasipneumoniae*	swine	128	0.06	2	2	>512	>512	>512	128	>256	256	128	>128	32	32	>128/64	32/16	128	128	0.5	0.5
T3	*K. pneumoniae*	swine	>128	0.125	2	2	>512	>512	>512	256	>256	256	>128	>128	0.125	0.5	>128/64	128/64	>128	>128	1	1
A19	*E. coli*	swine	128	0.032	4	4	>512	512	512	32	>256	256	8	16	0.5	0.5	>128/64	32/16	128	128	0.125	0.125
C45	*E. coli*	swine	128	0.032	2	2	>512	128	512	32	>256	64	>128	16	0.5	0.5	>128/64	16/8	128	128	0.125	0.125
D22	*E. coli*	swine	32	0.032	1	4	>512	256	512	32	256	128	>128	>128	32	16	128/64	16/8	128	128	0.5	0.5
0970	*E. coli*	companion animal	>128	0.125	2	2	>512	>512	>512	256	>256	>256	128	>128	>128	>128	>128/64	16/8	>128	>128	0.25	0.125
10194	*E. coli*	companion animal	128	0.125	4	4	>512	>512	>512	>256	>256	>256	128	>128	>128	>128	>128/64	>128/64	128	128	0.25	0.125
S51	*E. coli*	human	32	0.125	1	1	>512	4	512	0.06	256	1	4	4	1	1	>128/64	2/1	128	128	0.25	0.125
S69	*E. coli*	human	64	0.032	1	2	>512	1	512	0.03	256	0.5	2	2	1	1	128/64	2/1	128	>128	0.125	0.125
S87	*E. coli*	human	64	0.032	1	2	>512	>512	512	256	256	1	4	4	1	1	>128/64	16/8	128	128	0.25	0.125

+ indicates conjugants carrying the p3R-IncX3 plasmid; − indicates donors; MEM, meropenem; CT, colistin sulphate; AMP, ampicillin; CRO, ceftriaxone; EFT, ceftiofur; C, chloramphenicol; CN, gentamicin; AMC, amoxycillin/clavulanic acid; TET, tetracycline; TGC, tigecycline.

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
