# Peer review of "Fitness Cost of blaNDM-5-Carrying p3R-IncX3 Plasmids in Wild-Type NDM-Free Enterobacteriaceae"

_microorganisms, 2020, doi:10.3390/microorganisms8030377_

Round 1
Reviewer 1 Report
This is quite an interesting study on potential fitness cost of strained bearing a plasmid from the IncX3 group. Large number of strains was tested which is the major advantage of this work relative the previously published reports. There are, however, several problems with data interpretation and analysis, as well as presentation, which must be improved.
Major points:
(1) Results of experiments in which growth rates and biofilm formation abilities of tested strains are not properly analyzed. In Fig. 3, statistical analysis is mostly missing. Growth rates of all strains should be calculated on the basis of obtained results, and differences between plasmid-free and plasmids-containing strains should be tested for statistical significance. Then, differences should be considered only if statistical significance is reached, and other differences are just random fluctuations. This problem is even more pronouced in Fig. 4. Although statistical analysis is provided, the authors missinterpreted many results. For example, they stated (lines 277-278) that the DH5alpha strain expressed low ability to form biofilm, while differecnes between results of experiments with and without plasmid are statistically significant. Then, a decrease in this parameter in strain no. 15 is mentioned, while there is no statistical significance of the difference between both tested strains - this is just random fluctuation, not significant difference. Therefore, all the results presented in Fig. 3 and Fig. 4 should be re-analyzed, considering statistical analyses, and appropriate conclusions (based on appropriate pernetages of particular classes of strains) should be presented.
(2) For plasmid stability assessment, the described method was not appopriate. At each time point, the strains should be plated on both antibiotic-free and antibiotic-containing plates. Moreover, colonies should be tested by PCR not only at day 2, but preferably at each time point, and at least at also at day 15. Otherwise, it is unclar how the plasmid presence in bacteria could be determined.
Minor points:
(3) Lines 61-62: it is not true that plasmid loss can cause reduced growth rate. In fact, it is just opposite. Please, change this sentence.
(4) Line 66: "this study" does not make sense in the context. Did the authors mean "that study" or "those studies"?
(5) Line 89: Please, provide either a published article as the reference to the method (not web page which may be changed) or describe the method in more detail if the method was not published previously in the form of an article.
(6) The islated plasmid, from the IncX3 group, should be specifically named. Then, the authors can use the name "the IncX3 plasmid" in the text, but since there are different IncX3 plasmids, this one should be easily indentified.
(7) How was the strain DH5alpha/IncX3 constructed?
(8) The genomic sequences, determined in this work, should be deposited in a data base. Please, provide such information together with accesion numbers.
(9) Description of further research plans should be removed from Conclusions.
Reviewer 2 Report
Fitness cost of blaNDM-5-carrying IncX3 plasmids in. wild-type NDM-free Enterobacteriaceae. In this manuscript the. authors collected 41 wild type NDM-free Enterobacteriaceae isolates from different sources including the environment, chickens, meat, flies, humans, swine, and companion animals, to assess the success rate of conjugative transfer of a blaNDM-5-harboring IncX3 plasmid and then evaluated the fitness costs associated with plasmid acquisition. The manuscript is well written, and answer to an interesting question and the laboratory methods appear appropriate. There is no doubt on the originality of the study. The study provides interesting new data that could be useful for the comprehension and study of fitness cost associated. With plasmid acquisition.
The authors use established methods for sample collection, data analysis, and interpretation. No obvious errors were noted.
I recommend this manuscript "Accept as is".
My only minor point is that the authors need to thourghly discus the methods of selection of blaNDM- negative strains from their laboratory culture collection to be use as wild-type recipients
Round 2
Reviewer 1 Report
The authors have addressed all previous comments. In my opinion, the manuscript is acceptable now.